# Social inequities and clinical outcomes in young women with cervical cancer: Real-world evidence

**Luiza Cupertino Bérgomi**[1], **Giselle de Souza Carvalho**[1],
**Lucas Zanetti de Albuquerque**[1], **Daniel Gonçalves Kischinhevsky**[1],
**Alexssandra Lima Siqueira dos Santos**[1], **Luiza de Freitas Maciel**[2],
**Isabele Ávila Small**[1], **Jessé Lopes da Silva**[1‡], **Andreia C. de Melo**[1‡*]

**1** Division of Clinical Research and Technological Development, Brazilian National Cancer Institute, Rio de Janeiro, Brazil, **2** Gynecologic Oncology Section, Brazilian National Cancer Institute, Rio de Janeiro, Brazil

☯ These authors contributed equally to this study.
‡ These authors have contributed equally to this work.
* andreia.melo@inca.gov.br

## Abstract

### Objective

To assess the influence of social inequities and clinicopathological factors on survival outcomes in young women with cervical cancer treated at a comprehensive public cancer center in Brazil.

### Methods

This retrospective analysis reviewed the medical records of women aged 18–39 diagnosed with cervical cancer at the Brazilian National Cancer Institute between January 2017 and December 2021, assessing demographic characteristics and survival outcomes.

### Results

This analysis included 475 patients with a mean age of 33.6 years, with the majority being non-white (67.7%), never married women (68.0%), and having a low education level (< 8 years) (86.1%). Multivariate analysis indicated that a lower education level was associated with advanced stage (p = 0.001). Recurrence or progression occurred in 224 patients (47.2%), mainly as distant metastases (56.7%). The median progression-free survival (PFS) was 19.8 months, with two-year rates of 81.6%, 45.7%, 28.2%, and 6.2% for stages I, II, III, and IV, respectively. Shorter PFS was correlated with lower education level (p = 0.009), alcohol consumption (p = 0.026), undifferentiated carcinoma (p = 0.007), and advanced disease stage (p < 0.001). The median overall survival (OS) was 35.1 months, with five-year rates of 82.9%, 42.7%, 23.7%, and 9.7% for stages I, II, III, and IV, respectively. Factors associated with

**Data availability statement:** All relevant data are within the manuscript and its Supporting Information files.

**Funding:** The author(s) received no specific funding for this work.

**Competing interests:** ALSS has been paid for speaker's bureau from Astrazeneca, GSK, Bayer and Daichii, and has received support for attending meetings from Astrazeneca. JLS has been paid for speaker's bureau from MSD, Astrazeneca, Daichii, Pfizer and Roche, and has received support for attending meetings from Daichii and Astrazeneca. ACM has been paid for speaker's bureau from Astrazeneca, Bristol Myers Squibb, GSK, MSD, Novartis, Adium, Daichii and Pfizer and Roche and has received support for attending meetings from MSD, Daichii and Abbvie.

shorter OS included lower education level (p = 0.005), undifferentiated carcinoma (p = 0.006), and advanced stage (p < 0.001).

## Conclusion

Undifferentiated carcinomas and advanced stages negatively influence the prognosis of young women with cervical cancer. Social factors may also be correlated with poorer outcomes, especially alcohol consumption and lower education levels.

---

## Introduction

Cervical cancer represents a substantial burden on public health systems, particularly for women in Brazil. According to estimates from the Brazilian National Cancer Institute (INCA), cervical cancer is the third most common cancer among Brazilian women [1]. Furthermore, the global mortality rates for cervical cancer were reported to be 6.51 and 7.20 per 100,000 women in 2019 and 2020, respectively [2,3].

The most common histopathological subtypes of cervical cancer are squamous cell carcinoma and adenocarcinoma [4], with the former subtype being strongly linked to persistent Human Papillomavirus (HPV) infection and the development of cervical cancer within years [5]. As HPV-related cervical cancer typically develops after chronic infection, most diagnoses are expected to occur in older individuals. However, cervical cancer has been diagnosed in younger individuals, with some evidence indicating conflicting findings regarding age and clinical outcomes [6,7].

The diagnosis of gynecologic cancers profoundly influences the lives of female patients, often resulting in severe economic hardships for both them and their families, which can affect treatment adherence, health outcomes, and quality of life [8,9]. Notably, cervical cancer is recognized as a significant social determinant of health in underdeveloped countries, with studies showing that many patients come from low socioeconomic backgrounds and frequently lack access to adequate treatment. Poorer survival rates can be correlated with sociodemographic factors such as disease stage, rural residence, and employment status [10].

Exploring sociodemographic data to profile young patients with cervical cancer in Brazil is essential for identifying factors that influence health outcomes and advocating targeted prevention and early detection. This study aimed to detail the clinical and sociodemographic profiles of patients treated at the Brazilian National Cancer Institute (INCA), a public cancer center, and to assess the impact of these factors on survival outcomes.

## Materials and methods

This retrospective cohort study was approved by the Ethics in Human Research Committee of INCA, Rio de Janeiro, Brazil (registration number CAAE 73257023.5.0000.5274) and followed the Good Clinical Practice Guidelines and STROBE statement. Considering the retrospective nature of this observational study, the Institutional Review Board deemed it appropriate to waive the requirement for informed consent from participants.

The primary objective was to evaluate the outcomes, including progression-free survival (PFS) and overall survival (OS), of the young population treated at a public Brazilian institute. The secondary objective was to correlate sociodemographic and clinicopathological factors with these results.

A comprehensive analysis of the institutional database identified women aged 18–39 years who had been diagnosed with cervical cancer (adenocarcinoma, squamous cell carcinoma, or adenosquamous carcinoma) of any stage and were enrolled for treatment at the INCA between January 1, 2017, and December 31, 2021. Patients with unavailable source data, diagnosis of carcinoma in situ, initial treatment outside the INCA, and those with synchronous or metachronous tumors were excluded from the study. Data were collected from 01/12/2023 to 27/11/2024. The authors had access to information that could identify the participants while searching in the medical records. Despite this, anonymity was maintained and no identification was released from the statistical analysis and forward.

Relevant clinical information, including sociodemographic factors such as age, Eastern Cooperative Oncology Group performance status (ECOG PS), self-reported race/skin color based on the IBGE (Instituto Brasileiro de Geografia e Estatística) criteria [11], marital status, education level, distance from home to INCA, smoking habits, alcohol consumption, body mass index (BMI), comorbidities, histological type, tumor grading, staging, and treatment details, were retrieved retrospectively from medical records. Staging was performed according to the criteria established by the International Federation of Gynecology and Obstetrics (FIGO) 2018.

For the analysis of categorical variables, Pearson's chi-square test was employed, while continuous variables were assessed using Student's t-test, with missing data excluded from the analysis. PFS was determined from the date of diagnosis to the first occurrence of recurrence, as recorded in the institutional database, disease progression, death from any cause, or until the last follow-up for patients who did not experience disease progression. OS was defined as the time from diagnosis to death from any cause or the date of the last follow-up. PFS and OS estimates were derived using the Kaplan-Meier method. Hazard ratios (HR) were calculated using Cox proportional hazards models, with a multivariate analysis including all variables showing $p < 0.05$ in the univariate analysis. Statistical significance was set at $p < 0.05$. Statistical analyses were performed using R (version 3.5.3).

## Results

A total of 565 patients were selected for this study, of whom 475 met the eligibility criteria of this study. A flowchart of the exclusion criteria is shown in S1 Fig. The mean age was 33.6 years (Standard Deviation [SD] 4.3). Most patients were non-white (65.7%), single (68.0%), non-smokers (75.4%), and not consuming any alcohol (81.2%). The mean distance from the patients' homes to the cancer center was 37.8 km (SD 26.5). The mean BMI was 26.5 kg/m² (SD 5.8), with obesity present in 24% of the cases. Most women did not have comorbidities (83.1%). The most prevalent comorbidities were hypertension (9.5%) and Human Immunodeficiency Virus infection (4.2%). The mean number of sexual partners was 5.9 (SD 6), and the mean age at first sexual intercourse was 15.7 years (SD 2.2) (S1 Table).

The most common histological subtype was squamous cell carcinoma (81.3%). At diagnosis, 42.3% of patients had stage III/IV disease, with distant lymph nodes being the most common metastatic site (3.8%). In the multivariate analysis by stage (stage I/II *versus* stage III/IV), women with education < 8 years (47.5%, $p < 0.001$), squamous cell type (87.6%, $p = 0.001$), and ECOG PS 2–4 (27.8%, $p = 0.002$) were more prevalent in the stage III/IV subgroup (Table 1).

S2 Table summarizes the treatment data by stage at diagnosis. Notably, surgery was performed in 64% of stage I cases, with optimal resection (negative margins) achieved in 92% of cases, whereas chemoradiotherapy was administered in 89.1% of stage II cases and 61.4% of stage III cases. In contrast, 41.7% of stage IV cases were managed with palliative care. Cisplatin emerged as the predominant chemotherapy regimen in stages I, II and III, used in 95.7%, 99.2% and 97.9%, respectively. In stage IV, 52.9% of patients were treated with carboplatin plus paclitaxel. Although the analysis was not planned to discriminate between IVA (n = 22) and IVB (n = 26), it was probably because of the balance between the two stages. Moreover, in stage IV, chemotherapy was definitively discontinued in 58.8% of patients who did not meet the criteria for disease progression, primarily due to worsening ECOG PS (8.3%) and renal dysfunction (8.3%).

**Table 1. Univariate and multivariate analyses of variables at the initial staging of cervical cancer.**

| Variables | I-II (%) | III-IV (%) | Univariate analysis | | | Multivariate analysis | | |
|---|---|---|---|---|---|---|---|---|
| | | | OR | 95% CI | *p*-value | OR | 95% CI | *p*-value |
| **Mean age, years (SD)** | 33.5 (4.2) | 33.7 (4.4) | 1.01 | 0.97-1.06 | 0.627 | | | |
| **Ethnicity** | | | | | | | | |
| White | 96 (35.3) | 66 (33.0) | | | | | | |
| Non-white | 176 (64.7) | 134 (67.0) | 1.11 | 0.75-1.63 | 0.604 | | | |
| **Marital status** | | | | | | | | |
| Married | 82 (30.1) | 44 (22.0) | | | | | | |
| Not married | 190 (69.9) | 156 (78.0) | 1.53 | 1.01-2.35 | 0.049 | | | |
| **Education** | | | | | | | | |
| < 8 years | 82 (30.4) | 94 (47.5) | | | | | | |
| ≥ 8 years | 188 (69.6) | 104 (52.5) | 0.48 | 0.33-0.71 | <0.001 | 0.49 | 0.33-0.72 | <0.001 |
| **Smoking status** | | | | | | | | |
| No | 208 (78.8) | 130 (70.7) | | | | | | |
| Yes (former or current smoker) | 56 (21.2) | 54 (29.3) | 1.54 | 1.00-2.38 | 0.050 | | | |
| **Alcohol consumption** | | | | | | | | |
| No | 215 (83.7) | 140 (77.8) | | | | | | |
| Yes | 42 (16.3) | 40 (22.2) | 1.46 | 0.90-2.37 | 0.122 | | | |
| **Obese (≥ 30 kg/m²)** | | | | | | | | |
| No | 187 (72.5) | 140 (81.4) | | | | | | |
| Yes | 71 (27.5) | 32 (18.6) | 0.60 | 0.37-0.96 | 0.35 | | | |
| **ECOG PS** | | | | | | | | |
| 0 - 1 | 257 (97.3) | 135 (72.2) | | | | | | |
| 2 - 4 | 7 (2.7) | 52 (27.8) | 14.14 | 6.66-34.91 | <0.001 | 8.00 | 2.38-37.07 | 0.002 |
| **Comorbidities** | | | | | | | | |
| No | 221 (81.0) | 168 (86.2) | | | | | | |
| Yes | 52 (19.0) | 27 (13.8) | 0.68 | 0.41-1.12 | 0.140 | | | |
| **HIV-infection** | | | | | | | | |
| Yes | 6 (3.5) | 7 (5.1) | | | | | | |
| No | 165 (96.5) | 131 (94.9) | 0.68 | 0.21-2.10 | 0.498 | | | |
| **STI** | | | | | | | | |
| Yes | 5 (2.1) | 11 (6.5) | | | | | | |
| No | 228 (97.9) | 157 (93.5) | 0.31 | 0.10-0.88 | 0.034 | | | |
| **Histological type** | | | | | | | | |
| Squamous cell carcinoma | 210 (76.6) | 176 (87.6) | | | | | | |
| Adenocarcinoma / Adenosquamous | 62 (22.6) | 19 (9.5) | 0.37 | 0.21-0.62 | <0.001 | 0.38 | 0.21-0.65 | 0.001 |
| Undifferentiated carcinoma | 2 (0.7) | 6 (3.0) | 3.58 | 0.81-24.64 | 0.121 | 2.98 | 0.62-21.27 | 0.201 |
| **Tumor grade** | | | | | | | | |
| Grade 1 | 28 (12.0) | 7 (3.9) | | | | | | |
| Grade 2 | 138 (59.2) | 115 (64.6) | 3.33 | 1.48-8.54 | 0.006 | | | |
| Grade 3 | 67 (28.8) | 56 (31.5) | 3.34 | 1.42-8.84 | 0.009 | | | |

OR, Odds Ratio; SD, Standard Deviation; ECOG PS, Eastern Cooperative Oncology Group Performance Status; HIV, Human Immunodeficiency Virus; STI, Sexually Transmitted Infection.

The median follow-up was 59.1 months (95% CI: 56.0–61.4). The median PFS of the overall population was 19.8 months (95% CI: 17.1–24.8). Fig 1 A-C and S2 Fig A-C indicate the variables analyzed using the Kaplan-Meier method, with PFS being the most important. Notably, patients with education ≥ 8 years (HR 0.60; p < 0.001) and those with the adenocarcinoma subtype (HR 0.59; p = 0.003) experienced improved PFS outcomes. Conversely, alcohol consumption (HR 1.44; p = 0.016), undifferentiated carcinoma subtype (HR 4.44; p < 0.001), grade 3 tumors (HR 2.26; p = 0.003), and stage III/IV (HR 3.46; p < 0.001) were associated with poorer PFS. Multivariate analysis reaffirmed that women with education ≥ 8 years (HR 0.72; p = 0.009) showed significantly better PFS, while those with undifferentiated carcinoma subtype (HR 3.39; p = 0.007) and advanced disease stages (III/IV) (HR 2.95; p < 0.001) had adverse PFS (Table 2). Fig 2 displays a swimmer plot representing the individual patient's journey with stage IV from diagnosis to the last follow-up or death.

The median OS for the overall population was 35.3 months (95% CI: 28.2–51.4). OS by the Kaplan-Meier method according to the most expressed variables is shown in Fig 1D and 1F and S2 Fig D-F. Univariate analysis revealed that women with education ≥ 8 years (HR 0.59; p < 0.001) and those diagnosed with adenocarcinoma/adenosquamous subtypes (HR 0.50; p = 0.001) had significantly improved OS. Conversely, undifferentiated carcinoma subtype (HR 3.85; p < 0.001), grade 3 tumors (HR 2.42; p = 0.002), and advanced disease stages (III/IV) (HR 4.02; p < 0.001) were significantly associated with poorer OS. In the multivariate analysis, education ≥ 8 years (HR 0.69; p = 0.005) emerged as a positive prognostic factor of OS. In contrast, the undifferentiated carcinoma subtype (HR 3.14; p = 0.006) and stages III/IV (HR 3.80; p < 0.001) were associated with an increased risk of death (Table 3).

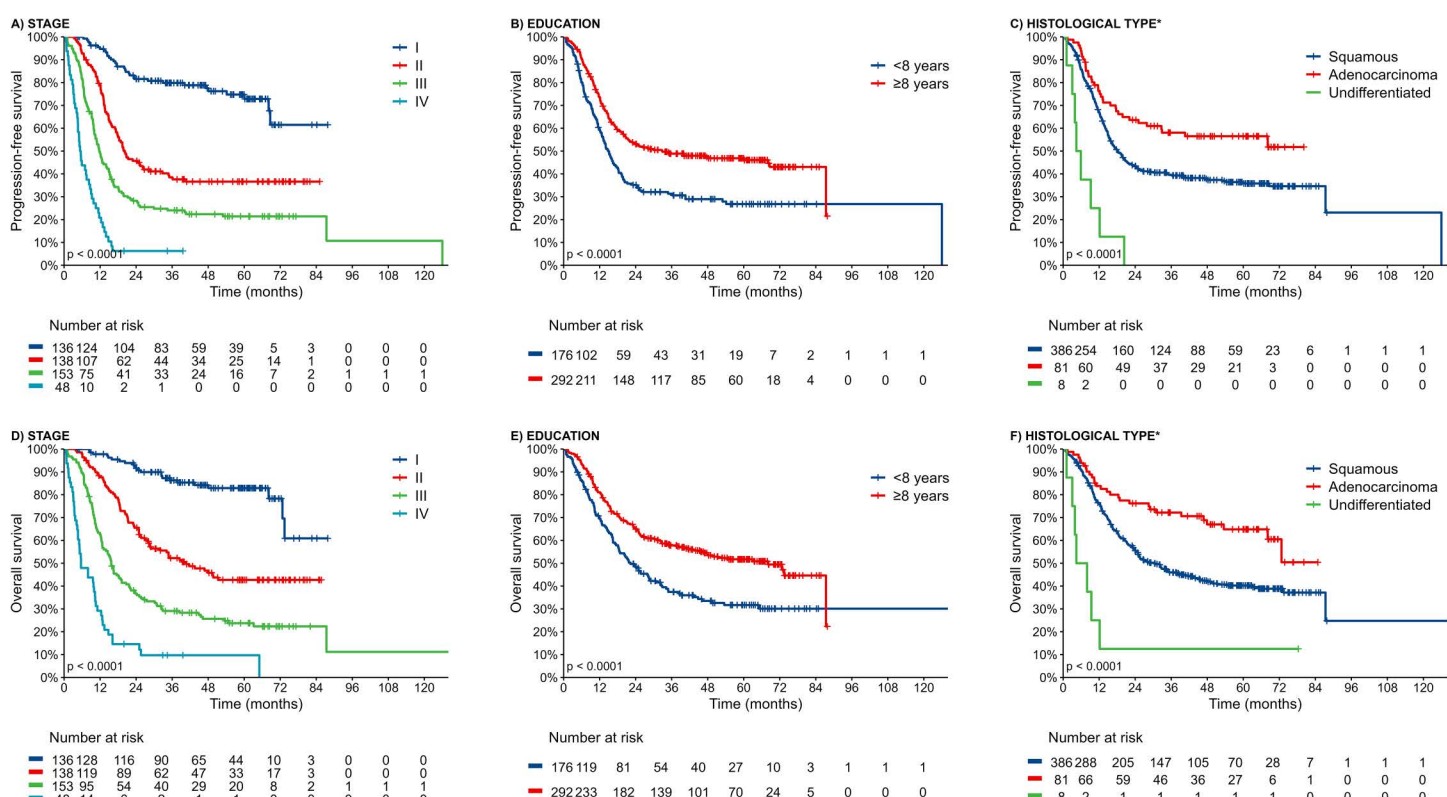

**Fig 1. Significant variables in Progression-free Survival and Overall Survival.** Progression-free Survival by A) Stage, B) Education, C) Histological type. Overall Survival by D) Stage, E) Education and F) Histological type. Thick marks indicate censored data. * Histological type: Squamous = Squamous cell carcinoma; Adenocarcinoma = adenocarcinoma and adenosquamous carcinoma; Undifferentiated = undifferentiated carcinoma.

**Table 2. Crude and adjusted hazard ratios for cervical cancer Progression-free Survival in univariate and multivariate analyses.**

| Variables | Univariate analysis | | | Multivariate analysis | | |
|---|---|---|---|---|---|---|
| | HR | 95% CI | *p*-value | HR | 95% CI | *p*-value |
| **Age** | 0.99 | 0.96-1.02 | 0.388 | | | |
| **Ethnicity** | | | | | | |
| White | | | | | | |
| Non-white | 1.00 | 0.78-1.28 | 0.998 | | | |
| **Marital status** | | | | | | |
| Married | | | | | | |
| Not married | 1.25 | 0.96-1.64 | 0.103 | | | |
| **Education** | | | | | | |
| < 8 years | | | | | | |
| ≥ 8 years | 0.60 | 0.47-0.76 | <0.001 | 0.72 | 0.56-0.92 | 0.009 |
| **Smoking** | | | | | | |
| No | | | | | | |
| Yes (former or current smoker) | 1.12 | 0.84-1.48 | 0.438 | | | |
| **Alcohol consumption** | | | | | | |
| No | | | | | | |
| Yes (former or current smoker) | 1.44 | 1.07-1.93 | 0.016 | 1.40 | 1.04-1.89 | 0.026 |
| **Obese (≥ 30 kg/m²)** | | | | | | |
| No | | | | | | |
| Yes | 0.96 | 0.72-1.28 | 0.771 | | | |
| **Histological type** | | | | | | |
| Squamous cell carcinoma | – | | | | | |
| Adenocarcinoma/ Adenosquamous | 0.59 | 0.41-0.84 | 0.003 | 0.73 | 0.49-1.10 | 0.132 |
| Undifferentiated carcinoma | 4.44 | 2.19-9.02 | <0.001 | 3.39 | 1.39-8.28 | 0.007 |
| **Tumor grade** | | | | | | |
| Grade 1 | | | | | | |
| Grade 2 | 1.42 | 0.85-2.38 | 0.185 | | | |
| Grade 3 | 2.26 | 1.33-3.85 | 0.003 | | | |
| **Stage** | | | | | | |
| I-II | | | | | | |
| III-IV | 3.46 | 2.72-4.40 | <0.001 | 2.95 | 2.27-3.83 | <0.001 |

HR, Hazard Ratio.

## Discussion

The large sample size enabled a comprehensive analysis of the adverse sociodemographic and clinicopathological profiles of young adults with cervical cancer in this cohort. Key findings revealed a high prevalence of non-white women, squamous cell carcinoma, single marital status, and multiple sexual partners, with many women living far from the cancer center. This cohort also showed early onset of sexual intercourse, multiparity, and increased rates of obesity, smoking, and alcohol consumption, along with advanced disease stage at diagnosis. Additionally, lower educational level, alcohol consumption, undifferentiated carcinoma subtype, and advanced stage were significant factors that negatively affected survival outcomes.

Most women were diagnosed with squamous cell carcinoma. Notably, undifferentiated carcinoma was the only histo-logical type significantly associated with poorer PFS and OS. Pan *et al.* found that adenocarcinoma did not significantly

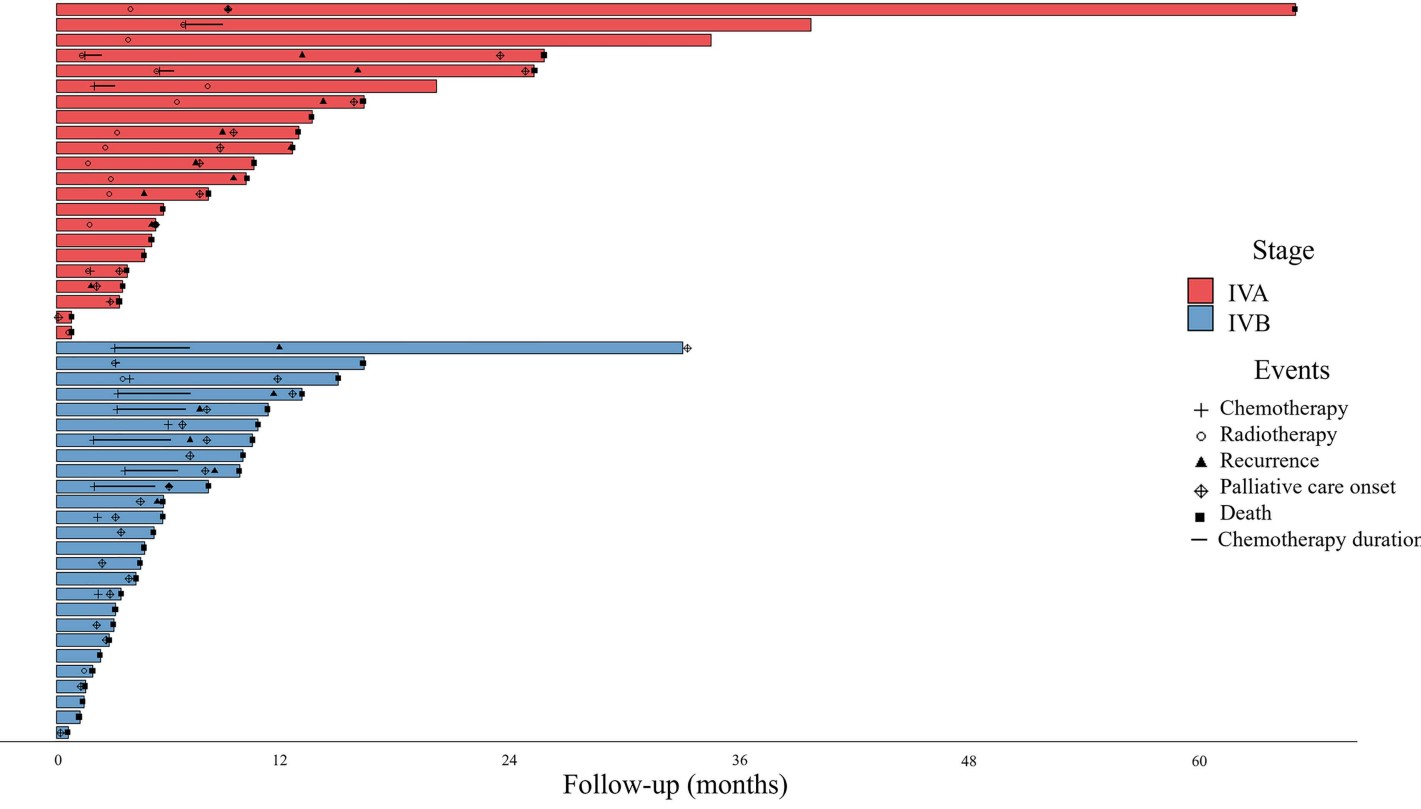

**Fig 2. Swimmer plot illustrating the individual patient's journey.**

impact mortality, whereas other histological types were associated with worse OS [12]. Pan *et al.* included rare histological types, such as small cell carcinoma, and the relatively small number of patients with undifferentiated carcinoma in the present cohort precludes definitive conclusions from this subgroup.

The INCA research team conducted a population-based study on cervical cancer incidence and mortality in Brazil, highlighting the alarming racial disparities faced by Black and Indigenous women. Between 2010 and 2015, Black women had a 44% higher risk of being diagnosed with cervical cancer than White women, while Indigenous women faced an even greater challenge, with an 82% increased mortality risk. The study revealed that systemic barriers to healthcare access are exacerbated by structural racism, emphasizing the need for targeted interventions such as improved HPV vaccination campaigns and effective screening strategies [13]. Similar trends have been reported in other countries, where racial and ethnic minorities encounter significant barriers to treatment, resulting in lower survival rates [14]. In the United States, for example, studies have shown that Black women are often diagnosed at more advanced stages and are less likely to receive timely and appropriate treatments [15].

In this study, the mean distance from home to INCA was 37.8 km, reflecting its importance as a regional reference cancer center for the surrounding cities. Stage III cervical cancer was the most common diagnosis, and most patients received chemoradiotherapy. Although most patients received appropriate treatment, a significant proportion of young women were diagnosed with *de novo* stage IV disease and were directly referred to palliative care, possibly due to clinical impairment. The reasons for the absence of systemic treatment were not systematically recorded, as this significant amount was not expected to be observed. One explanation could be the evolution of ineligible ECOG PS for chemotherapy while awaiting referral to an oncology center.

**Table 3. Crude and adjusted hazard ratios for cervical cancer Overall Survival in univariate and multivariate analyses.**

| Variables | Univariate analysis | | | Multivariate analysis | | |
|---|---|---|---|---|---|---|
| | HR | 95% CI | *p*-value | HR | 95% CI | *p*-value |
| **Age** | 0.99 | 0.96-1.02 | 0.397 | 0.97 | 0.94-1.00 | 0.027 |
| **Marital status** | | | | | | |
| Married | | | | | | |
| Not married | 1.40 | 1.04-1.87 | 0.024 | | | |
| **Education** | | | | | | |
| < 8 years | | | | | | |
| ≥ 8 years | 0.59 | 0.46-0.76 | <0.001 | 0.69 | 0.54-0.89 | 0.005 |
| **Smoking** | | | | | | |
| No | | | | | | |
| Yes (former or current smoker) | 1.20 | 0.90-1.61 | 0.211 | | | |
| **Alcohol consumption** | | | | | | |
| No | | | | | | |
| Yes | 1.42 | 1.04-1.94 | 0.026 | | | |
| **Histological type** | | | | | | |
| Squamous cell carcinoma | | | | | | |
| Adenocarcinoma / Adenosquamous | 0.50 | 0.34-0.74 | 0.001 | 0.77 | 0.52-1.16 | 0.217 |
| Undifferentiated carcinoma | 3.85 | 1.80-8.20 | <0.001 | 3.14 | 1.39-7.14 | 0.006 |
| **Tumor grade** | | | | | | |
| Grade 1 | | | | | | |
| Grade 2 | 1.46 | 0.84-2.53 | 0.180 | | | |
| Grade 3 | 2.42 | 1.37-4.27 | 0.002 | | | |
| **Stage** | | | | | | |
| I-II | | | | | | |
| III-IV | 4.02 | 3.12-5.20 | <0.001 | 3.80 | 2.91-4.98 | <0.001 |

HR, Hazard Ratio.

This study found that higher educational levels were associated with better survival outcomes. Vincerževskienė *et al.* reported poorer outcomes for patients with < 9 years of formal education, although this association did not reach statistical significance, and participants were aged between 25 and 64 years [16]. Additionally, an Indian study indicated that while higher education improved OS, this relationship lost statistical significance after adjusting for disease stage [17]. Lower educational levels may correlate with barriers to healthcare access, transportation issues and increased financial responsibilities.

Alcohol consumption emerged as a statistically significant risk factor associated with shorter PFS, although it did not show a significant association with OS. Mayadev *et al.* conducted a comprehensive study investigating the impact of alcohol abuse on pelvic control and survival outcomes in patients with locally advanced cervical cancer receiving radiation treatment. Their findings indicated that heavy alcohol consumption was significantly correlated with reduced disease-free survival and OS rates in the studied population. Specifically, the study highlighted that approximately 10.5% of patients were classified as heavy drinkers, and their alcohol use adversely affected treatment efficacy and led to a higher recurrence rate [18]. Furthermore, higher cervical cancer incidence and mortality rates have been reported in countries with high alcohol consumption levels [19].

Another important finding of this study was that nearly half of the patients experienced recurrence or disease progression. Distant recurrence occurred in 56.7% of cases, primarily affecting the lymph nodes. Previous reports have indicated

distant recurrence rates ranging from 24.2% to 33.3%, depending on the initial disease stage and risk factors [20,21]. Kim *et al.* reported a predominant distant recurrence rate of 59.5%, particularly in lymph nodes [22].

Although several studies have focused on young patients, differences in the definition of young age hinder direct comparison. For instance, one study defined young patients as those under 25 years of age and reported a 5-year OS rate of 83.7%, with 71.6% diagnosed with stage I disease [12]. Another Brazilian study, also with young patients under 25 years of age, reported a 5-year OS rate of 60% [23]. Furthermore, the analysis of young versus older patients may have the same obstacles as the definition of age itself. Barben *et al.* compared young women (<70 years) with older women (≥70 years), and reported that the 5-year survival rates were 74.1% and 36.4%, respectively [6].

As for survival comparisons between young women and the overall population, the results may differ according to lower- or higher-income countries. This entire cohort of young patients demonstrated a median OS and 5-year survival rate of 35.3 months and 44.0%, respectively. Tabatabaei *et al.* studied a population of 187 patients in an academic referral cancer center in Iran and found a median OS of 24 months [24]. However, a German study conducted by Chen *et al.* described a 5-year relative survival rate of 64.7% [25].

This study had several strengths, particularly its clearly defined population. Conducted at a high-volume public facility with standardized treatment protocols, this study minimized the variability in clinical practice. The large sample size further increased the statistical power, enabling robust identification of associations and possible generalization of the findings to similar populations. In addition, it is the largest study to focus on patients with cervical cancer under 40 years of age.

The limitations include its retrospective design, which restricts causal inference and may introduce biases such as selection bias or misclassification of exposures and outcomes based on historical record quality. Moreover, the absence of a comparison group of older patients limits the analyses of prior studies.

Cervical cancer poses a significant health challenge for young women in Brazil. The results demonstrate that sociodemographic factors may be correlated with worse outcomes. This study calls for targeted public health initiatives to raise awareness, promote early screening, and ensure equitable access to treatment.

## Conclusion

This study highlights the effects of sociodemographic factors, histological type, and stage of cervical cancer on the prognosis in young women. Undifferentiated tumors and advanced-stage disease were associated with poor outcomes. Additionally, lower education levels and alcohol consumption increased the risk of adverse effects in this study. These findings may aid in the development of public implementation programs for young women.

## Supporting information

**S1 Fig. Flowchart of patient screening, eligibility and inclusion.**
(TIF)

**S2 Fig. Progression-free survival and Overall Survival variants.** Progression-free survival by A) Overall population, B) Alcohol consumption, C) Tumor grade. Overall survival by D) Overall population, E) Alcohol consumption, F) Tumor grade. Thick marks indicate censored data.
(TIF)

**S1 Table. Clinical and histopathological characteristics for the overall population.** SD, Standard Deviation; BMI, Body Mass Index.
(DOCX)

**S2 Table. Overview of treatment data by stage.** CRT, Chemoradiotherapy; RT, Radiotherapy; CT, Chemotherapy; CDDP, cisplatin; CP, carboplatin-paclitaxel; IQR, Interquartile Range; PS, Performance Status; BT, Brachytherapy.
(DOCX)

## Acknowledgments

The authors express their profound gratitude to all patients who participated in this study.

## Author contributions

**Conceptualization:** Luiza Cupertino Bérgomi, Giselle de Souza Carvalho, Lucas Zanetti de Albuquerque, Daniel Gonçalves Kischinhevsky, Alexssandra Lima Siqueira dos Santos, Luiza de Freitas Maciel, Isabele Ávila Small, Jessé Lopes da Silva, Andreia C. de Melo.

**Data curation:** Luiza Cupertino Bérgomi, Giselle de Souza Carvalho, Lucas Zanetti de Albuquerque, Daniel Gonçalves Kischinhevsky, Alexssandra Lima Siqueira dos Santos, Luiza de Freitas Maciel.

**Formal analysis:** Isabele Ávila Small.

**Methodology:** Luiza Cupertino Bérgomi, Giselle de Souza Carvalho, Lucas Zanetti de Albuquerque, Daniel Gonçalves Kischinhevsky, Alexssandra Lima Siqueira dos Santos, Luiza de Freitas Maciel, Isabele Ávila Small, Jessé Lopes da Silva, Andreia C. de Melo.

**Supervision:** Isabele Ávila Small, Jessé Lopes da Silva, Andreia C. de Melo.

**Writing – original draft:** Luiza Cupertino Bérgomi, Giselle de Souza Carvalho, Lucas Zanetti de Albuquerque, Daniel Gonçalves Kischinhevsky, Alexssandra Lima Siqueira dos Santos, Luiza de Freitas Maciel.

**Writing – review & editing:** Jessé Lopes da Silva, Andreia C. de Melo.

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
