## [Decision Letter · Decision Letter 0]

19 Jan 2026

Dear Dr. de Melo,

Thank you for submitting your manuscript to PLOS ONE. After careful consideration, we feel that it has merit but does not fully meet PLOS ONE’s publication criteria as it currently stands. Therefore, we invite you to submit a revised version of the manuscript that addresses the points raised during the review process.

We look forward to receiving your revised manuscript.

Kind regards,

Satyajeet Rath

Academic Editor

PLOS One

Journal Requirements:

2. We noted in your submission details that a portion of your manuscript may have been presented or published elsewhere. “The initial abstract was published as online version in ASCO 2025. The developed manuscript with figure, tables and in-depth discussion of the results is presented in this article.” Please clarify whether this was peer-reviewed and formally published. If this work was previously peer-reviewed and published, in the cover letter please provide the reason that this work does not constitute dual publication and should be included in the current manuscript.

Additional Editor Comments:

Kindly review the manuscript for certain grammatical mistakes. Answer the queries by the reviewers.

Additional comments: 1. Why was FIGO 2009 stage used in a study conducted in a sample from 2024-2025.

2. What novelty/value this study adds to the already existing known literature.

Reviewers' comments:

Reviewer's Responses to Questions

**Comments to the Author**

1. Is the manuscript technically sound, and do the data support the conclusions?

Reviewer #1: Yes

Reviewer #2: Yes

2. Has the statistical analysis been performed appropriately and rigorously?

Reviewer #1: Yes

Reviewer #2: Yes

3. Have the authors made all data underlying the findings in their manuscript fully available?

Reviewer #1: Yes

Reviewer #2: Yes

4. Is the manuscript presented in an intelligible fashion and written in standard English?

Reviewer #1: Yes

Reviewer #2: Yes

Reviewer #1: The manuscript from L. Cupertino Bérgomi et al. is clearly written. Data are presented in a linear and comprehensive fashion. Conclusions are supported by the data. I, therefore, believe that it is suitable for publication in PLOS One in its present form.

Reviewer #2: Review of article PONE-D-25-61883

The article “Social inequities and clinical outcome in young women with cervical cancer: real-world evidence” is well-written, organized and provides good analysis of the study findings. The statistical analyses used are simple yet effective. Some minor comments to consider:

1. The analysis was restricted to women age 18-39 years. Given the prevalence of cancer in older women, why was the analysis restricted to this age group?

2. Line 56 – suggest changing the language to ‘has been’ rather than ‘can also be’…

3. Line 100 – please define the ‘first occurrence of recurrence’ to that limited to what is recorded in the database only

4. Line 113 – please define ‘non-alcoholics” of replace with ‘not consuming any alcohol’.

**Do you want your identity to be public for this peer review?** For information about this choice, including consent withdrawal, please see our Privacy Policy

Reviewer #1: No

Reviewer #2: No

---

## [Author Response · Author response to Decision Letter 1]

29 Jan 2026

It is a great pleasure to be considered for publication in PLOS One. We appreciate the constructive suggestions, so we could use them to enhance our study. This letter has the intention to answer the queries by the reviewers. A detailed response to the questions is addressed below:

### Journal Requirements:

1. Please ensure that your manuscript meets PLOS ONE style requirements, including those for file naming. The PLOS ONE style templates can be found at:

https://journals.plos.org/plosone/s/fileid=wjVg/PLOSOne_formatting_sample_main_body.pdf and

Answer: Checked.

2. We noted in your submission details that a portion of your manuscript may have been presented or published elsewhere. “The initial abstract was published as online version in ASCO 2025. The developed manuscript with figure, tables and in-depth discussion of the results is presented in this article.” Please clarify whether this was peer-reviewed and formally published. If this work was previously peer-reviewed and published, in the cover letter please provide the reason that this work does not constitute dual publication and should be included in the current manuscript.

Answer: This study was selected for presentation as an online-only abstract at the ASCO Annual Meeting 2025. As is standard practice at ASCO, online-only abstracts undergo editorial screening for relevance and scientific merit, but do not constitute full peer-reviewed publications, nor do they include detailed methodological appraisal, complete statistical analyses, figures, or tables. The abstract was published solely in abstract form within the ASCO Meeting Proceedings.

At that time, only preliminary and summarized results were presented, without comprehensive and detailed survival analyses or stratified evaluations of sociodemographic determinants. Importantly, no figures, tables, extended methods, or in-depth interpretation were included in the ASCO abstract.

The current submission represents a substantially expanded and original full-length manuscript, with major additions that were not previously disseminated. These include a refined and complete statistical analysis, fully developed Kaplan–Meier and Cox regression models, detailed figures and tables, and an extensive discussion contextualizing the findings within the existing literature. Moreover, the manuscript advances the work conceptually by integrating social determinants of health into prognostic modeling and by explicitly discussing implications for patient care pathways and public health policy.

In accordance with ICMJE recommendations and the policies of major biomedical journals, prior presentation of preliminary data in abstract form at a scientific meeting does not constitute prior publication, nor does it represent duplicate or redundant publication. The present manuscript, therefore, constitutes the first complete, peer-reviewable, and citable report of this study.

Answer: Please refer to the previous answer.

### Additional Editor Comments:

Kindly review the manuscript for certain grammatical mistakes.

Answer the queries by the reviewers.

Answer: The complete manuscript was reviewed and edited to ensure clarity, consistency, and adherence to academic standards.

### Reviewers´comments:

Reviewer #1:

The manuscript from L. Cupertino Bérgomi et al. is clearly written. Data are presented in a linear and comprehensive fashion. Conclusions are supported by the data. I, therefore, believe that it is suitable for publication in PLOS One in its present form.

Answer: We appreciate that.

Reviewer #2: Review of article PONE-D-25-61883

The article “Social inequities and clinical outcome in young women with cervical cancer: real-world evidence” is well-written, organized and provides good analysis of the study findings. The statistical analyses used are simple yet effective. Some minor comments to consider:

1. The analysis was restricted to women age 18-39 years. Given the prevalence of

cancer in older women, why was the analysis restricted to this age group?

Answer: We thank the reviewer for this important question. The decision to restrict the analysis to women aged 18–39 years was deliberate and hypothesis-driven.

First, although cervical cancer remains more prevalent in older women, young women constitute a biologically, socially, and clinically distinct population. This age group is characterized by unique reproductive considerations, different patterns of HPV exposure and persistence, treatment-related fertility implications, and specific long-term survivorship and socioeconomic consequences. Pooling younger and older patients may therefore obscure age-specific associations and dilute the impact of social determinants on outcomes.

Second, emerging epidemiologic signals in Brazil motivated this focused analysis. While not yet fully published, ongoing analyses conducted by our research group using national population-based registries suggest a relative increase in cervical cancer diagnoses among younger women, particularly in socioeconomically vulnerable populations. These observations are consistent with international concerns regarding delayed screening uptake, inequitable access to preventive services, and heterogeneous implementation of HPV vaccination in low- and middle-income settings. The present study was designed to address this knowledge gap by providing detailed real-world evidence in a population for which robust data remain scarce.

Third, the literature addressing cervical cancer in young women is limited and heterogeneous, with inconsistent age cutoffs and conflicting results regarding prognosis and risk factors. By focusing on a clearly defined age range, this study contributes novel, granular data on clinicopathological features, survival outcomes, and social inequities in a population that is often underrepresented or merged into broader age-based analyses.

Finally, from a public health perspective, studying cervical cancer in young women has direct policy relevance. Disease occurring at younger ages carries disproportionate societal impact due to loss of productivity, caregiving responsibilities, and long-term survivorship burden. Identifying modifiable sociodemographic factors in this group may inform targeted prevention strategies, optimized screening policies, and equity-oriented interventions.

2. Line 56 – suggest changing the language to ‘has been’ rather than ‘can also be’…

Answer: Checked. Correction made.

3. Line 100 – please define the ‘first occurrence of recurrence’ to that limited to what

is recorded in the database only

Answer: Checked. Correction made.

4. Line 113 – please define ‘non-alcoholics” or replace with ‘not consuming any

alcohol’.

Answer: Checked. Correction made.

We thank the reviewers and editorial team once again for their insights.

With sincere regards,

---

## [Editor Report · Decision Letter 1]

9 Feb 2026

Social inequities and clinical outcomes in young women with cervical cancer: real-world evidence

PONE-D-25-61883R1

Dear Dr. de Melo,

We’re pleased to inform you that your manuscript has been judged scientifically suitable for publication and will be formally accepted for publication once it meets all outstanding technical requirements.

Kind regards,

Satyajeet Rath

Academic Editor

PLOS One

Additional Editor Comments (optional):

I commend the authors on making the required changes.
---

## [Editor Report · Acceptance letter]

PONE-D-25-61883R1

PLOS One

Dear Dr. de Melo,

I'm pleased to inform you that your manuscript has been deemed suitable for publication in PLOS One. Congratulations! Your manuscript is now being handed over to our production team.

Kind regards,

on behalf of

Dr. Satyajeet Rath

Academic Editor

PLOS One